# Analysis of Healthcare Expenditures in Bulgaria

**DOI:** 10.3390/healthcare10020274

**Published:** 2022-01-30

**Authors:** Zornitsa Mitkova, Miglena Doneva, Nikolay Gerasimov, Konstantin Tachkov, Maria Dimitrova, Maria Kamusheva, Guenka Petrova

**Affiliations:** 1Department of Organization and Economy of Pharmacy, Faculty of Pharmacy, Medical University-Sofia, 1000 Sofia, Bulgaria; miglena_doneva@abv.bg (M.D.); tachkov@outlook.com (K.T.); mia_dimitrova@yahoo.com (M.D.); maria.kamusheva@yahoo.com (M.K.); guenka.petrova@gmail.com (G.P.); 2Medical College, Trakia University, 6000 Stara Zagora, Bulgaria; nik.gerasimov.phd@gmail.com

**Keywords:** public expenditure, healthcare system performance, demographics, economic indicators

## Abstract

The growth of public expenditure worldwide has set the priority on assessment of trends and establishment of factors which generate the most significant public costs. The goal of the current study is to review the tendencies in public healthcare expenditures in Bulgaria and to analyze the influence of the demographic, economic, and healthcare system capacity indicators on expenditures dynamics. A retrospective, top-down, financial analysis of the healthcare system expenditures was performed. Datasets of the National Statistical Institute (NSI), National Health Insurance Fund (NHIF), and National Center of Public Health and Analysis (NCPHA) were retrospectively reviewed from2014–2019 to collect the information in absolute units of healthcare expenditures, healthcare system performance, demographics, and economic indicators. The research showed that increasing GDP led to higher healthcare costs, and it was the main factor affecting the cost growth in Bulgaria. The number of hospitalized patients and citizens in retirement age remained constant, confirming that their impact on healthcare costs was negligible. In conclusion, the population aging, average life expectancy, patient morbidity, and hospitalization rate altogether impacted healthcare costs mainly due to the multimorbidity of older people and the rising need for outpatient hospital services and medications.

## 1. Introduction

EU countries spent on average 9.6% of their gross domestic product (GDP) on healthcare in 2017 and 9.92% in 2019 [1,2]. Germany, France, and Sweden reported the highest healthcare expenditures in 2018 (between 10.9% and 11.5% of GDP). The largest growth in healthcare costs per inhabitant between 2012 and 2018 were observed in Romania and the Baltic Member States [2]. The USA spent 19.7% of its GDP on healthcare in 2019 [3] with a projected respective growth of 5.4%and 2.4% for medical goods and services annually for the period 2019–2028.

The largest share of healthcare costs is spent on noncommunicable diseases. In 27 EU countries and the UK, the direct costs of noncommunicable diseases (NCDs) are expected to increase by 0.8% annually from2014–2050, with population aging remaining the substantial driver of total healthcare expenditures [4].

There is a strong relationship among the level of income, country economic development, and population expenditures on healthcare [1]. To overcome the growth in healthcare expenditures, different types of cost containment measures are implemented across countries. A study in the USA found a positive correlation between healthcare expenditure and economic indicators such as income, GDP, and labor productivity. The authors considered that further investment in various healthcare aspects would turn to an improvement in income and GDP, as well as reduce the population poverty [5]. Healthcare expenditures can be affected by industrialization, urban population, and economic growth [6]. A developed model investigated the factors determining government expenditure on healthcare and found that expenditures are significantly affected by variables related to government revenues, population, and government debt [7].

There is evidence that price and reimbursement control are not associated with the cost containment of national health expenditures, while value-based pricing (VBP) of medicines may present a more effective mechanism in a long-term period [8]. Involvement of all principal healthcare stakeholders and the introduction of incentive-compatible remuneration schemes are expected to play a significant role in the future. Setting the budgetary targets improves cost–benefit considerations, and their assessment could be based on age-related morbidity and progress in medicines and their development [9].

The WHO report highlighted that all countries have to control health spending and outcomes, as well as identify the measures and actions considering the negative impact of economic decline [10].

In addition, the pandemic has had an adverse impact on health and healthcare systems, forcing or requiring a reorganization of funds and services [11]. The fact remains that central and eastern European countries spend less on healthcare in comparison to western countries, and that the structure of expenditures differs across countries [12,13].

The examination and assessment of tendencies in healthcare expenditures and factors affecting them can help in further cost control and optimization, as well as improve the efficiency and effectiveness of spending, as it is crucial to improve health and provide necessary health services to the population [14].

The goal of the current study was to review the tendencies in public healthcare expenditures in Bulgaria and factors affecting them.

The study is arranged in five parts in order to highlight the impact of each group of indicators on healthcare costs. First, we analyze the overall healthcare expenditure during 2014–2019. In the next three parts, we illustrate the dynamics and tendencies considering healthcare system performance indicators, demographics, and economic factors in the country. Additionally, their linkage is explored using graphs and statistical analysis in order to confirm the most significant factors affecting healthcare expenditure deviations in the country.

## 2. Materials and Methods

A retrospective, top-down, financial analysis of the healthcare system expenditures was performed. Data were collected from the datasets of the National Statistical Institute (NSI), National Health Insurance Fund (NHIF), and National Center of Public Health and Analysis (NCPHA). The officially published information covers healthcare expenditures, healthcare system performance, demographic, and economic indicators (Table 1), which were extracted retrospectively and analyzed for the period 2014–2019.

The study concept is based on the examination of the three main groups of factors affecting public healthcare expenditure, namely, demographic characteristics, economic factors, and healthcare system performance. The factors were selected on the basis of the literature, as they were the most common among systems [15,16,17,18], according to their significance for public healthcare expenditure performance. The total number of hospitalized patients, population over working age (retirement age), and GDP by final consumption expenditure were also extracted and included in the analysis.

**Table 1 healthcare-10-00274-t001:** Sources used for data extraction.

Group of Indicators	Observed Indicators	Sources
Public healthcare expenditure	→ Total cost for outpatient services. → Total cost for hospital services. → Total cost for retail trade, drugstores, optical stores, and other services.	National Statistical Institute: system of health accounts [19].
	→ NHIF expenditures for medical devices, medicines, and dietary foods. → NHIF expenditures for medicinal products.	NHIF annual reports published during 2014–2019 [20].The reports for the period 2014–2019 were observed.
Healthcare system performance indicators	→ Total number of hospitalized patients.	National Statistical Institute: healthcare [21].The reports for the period 2014–2019 were observed.
	→ Number of hospitalized patients due to diseases of the circulatory system. → Number of hospitalized patients due to cancer diseases. → Number of hospitalized patients due to respiratory system diseases.	National Center of Public Health and Analysis: healthcare short statistical guide [22].The reports for the period 2014–2019 were observed.
	→ Mortality rate. → Neonatal mortality rate.	National Statistical Institute: deaths and mortality by causes [23].
Demographic characteristics of population	→ Average life expectancy. → Population size. → Ratio of the population aged 65 and over. → Population over working age (retirement age).	National Statistical Institute: population—demography, migration, and forecasts [24].The reports for the period 2014–2019 were observed.
Economic indicators	→ GDP by final consumption expenditure, BGN. → GDP per capita BGN. → Average monetary expenditures per person. → Average monetary expenditures for healthcare per person. → Gross monetary income per person.	National Statistical Institute: macroeconomic statistics, GDP.
	→ Average yearly inflation in healthcare sector (%). → Inflation of medicinal products prices (%). → Inflation of medicinal devices and equipment prices (%).	The World Bank Data on inflation customer prices [25].

Relative indicators were calculated as per patient expenditures and yearly cost indices to analyze the tendencies in the expenditures and other indicators.

The annual percentage increase in healthcare expenditures and other factors were calculated. Using 2014 values as a basis, we calculated the percentage change in observed parameters during the period 2014–2019. Overall, the presented data graphically illustrated how the selected factors for each group correlate with healthcare expenditure.

A nonparametric Mann–Whitney test was applied to test statistically significant differences between healthcare expenditures during the observed period. Comparison of means was applied using Med Calc. A *p*-value lower than 0.05 is considered statistically significant. The average values of parameters and their standard deviations (SD) were calculated for the purposes of comparison.

All costs are presented in national currency BGN at the fixed for the period exchange rate of 1 BGN = 0.95 EUR.

## 3. Results

### 3.1. Overview of Health Expenditures in Bulgaria

Table 2 presents the healthcare expenditures during 2014–2019. There was a permanent growth in all observed indicators for the period. The cost for hospital services and cost for retail trade, drugstores, optical stores, and other services exhibited the highest value, and an increase was observed in all sectors.

About a two fold increase in average expenditures for hospital services was observed (2781.2 ± 320 million BGN vs. 1153 ± 101 million BGN). Its comparison with the outpatient costs indicates that healthcare services were predominantly hospital-orientated, indicating that more expenditures were focused on hospital care services. On the other hand, the cost for hospital and outpatient services exceeded the cost paid by the NHIF for medicinal products by factors of four and two, respectively (Table 2).

Another observation is that reimbursement coverage, i.e., the reimbursed part of the cost for medicines (average value for the period 714.5 ± 97 million BGN), was approximately four times lower than the overall cost for retail trade in drugstores, optical stores, and other services. This indicates that the burden for medicines was heavier on the patients (average value for the period 2759.5 ± 279 million BGN). Furthermore, the costs for medical devices and dietary foods reimbursed by the NHIF did not increase much, contrary to the reimbursed cost for medicinal products. We have to note that a limited number of medical devices and dietary foods are reimbursed. It is also worth mentioning that hospital and outpatient services are reimbursed by the NHIF, i.e., public financing is responsible for reimbursing the cost of hospital services.

Statistical analysis revealed statistically significant differences between healthcare expenditures for outpatient services and those for hospital services (*p*-value = 0.00512), but there were no statistically significant differences between public funding of medical devices, medicinal products, and dietary foods compared with the financing of medicinal products (*p*-value = 0.37886) during the observed period.

Per patient costs showed similar tendencies to those observed in the overall expenditures (Figure 1). One-quarter of the pharmaceutical market was reimbursed; however, only 3–4 BGN (1.5–2 EUR) was the amount reimbursed for medical devices and dietary food per patient. The cost for reimbursed hospital services per patient was three times higher than that for ambulatory care.

The mean changes in expenditures for the investigated period were between 4.5% and 6.8%. Changes in expenditures also fluctuated, and not all years saw notable increases. For some years, we noticed a decrease in expenditures (Table 3); however, the overall trend was upward. During the first 3 years of observations, the most notable increases were for the reimbursed costs of medicines; however, in the last 2 years, they decreased in comparison to the previous year, probably due to cost containment measures imposed by the NHIF. On the other hand, the total cost for retail trade increased, continuously highlighting that the burden for patient was increasing. The costs for outpatient and hospital services were increasing annually, albeit at different rates during 2015–2019, with 9.8% to 6.2% at the highest observed rate. There were no statistically significant differences between mean value of indices for NHIF healthcare expenditures for medicinal products and hospital services (*p*-value = 0.64), as well as between mean values of indices for total costs for outpatient services and mean values of total costs for retail trade, drugstores, optical stores, and other services (*p*-value = 0.795) during the observed period.

### 3.2. Healthcare System Performance Indicators

Examination of the indicators for healthcare system performance provided additional information regarding the main issues, leading to higher budget costs and rising expenditure for medicinal products and hospital services. Hospitalized patients remained almost constant at 2.2 million per year (Table 4). Three main groups of diseases (cardiovascular, respiratory, and oncology) were responsible for more than one-third of all hospitalizations. The number of hospitalized patients due to the first two groups was also quite constant, while hospitalized oncology patients permanently increased. On the basis of these two indicators, we can suppose that the increase in hospital expenditures was either due to the introduction of new health technologies in oncology or due to the increase in reimbursed cost for hospital services.

Mortality from cardiovascular diseases decreased, suggesting a probable improvement in their therapy, but that of oncology diseases slightly increased. Infant mortality is an important indicator when considering healthcare sector development and organization. Its decrease suggests an improvement in maternal healthcare and investments in new technologies.

### 3.3. Other Factors Affecting Healthcare Expenditure

To further examine the probable factors influencing the healthcare expenditures and sustainability of the healthcare system, we analyzed the demographics and some macroeconomic indicators (Table 5 and Table 6).

### 3.4. Demographic Indicators

The Bulgarian population steadily decreased, with a negative birth rate of (−0.7%) per year and stable average life expectancy during 2014–2019 (Table 5). The population over working age also slowly decreased, but its ratio as a percentage of the general population increased by 3% for the period. The aging population is usually one of the major consumers of healthcare services due to the development of chronic diseases, and it could increase the financial pressure on the system.

### 3.5. Economic Factors

A permanent growth of gross domestic product was observed (from 83 to 120 billion BGN) during 2014–2019 (Table 6). This was a positive reflection of the growing GDP per capita, as well as per person expenditures for goods and healthcare services. The income of the average person increased and, in combination with negative or small inflation rates, allowed for an increase in personal expenditures. Juxtaposing these economic factors alongside the healthcare indicators, we can assume that the major healthcare expenditure drivers were new health technologies and morbidity increases for some chronic diseases.

### 3.6. The Relationship between Healthcare Expenditure and Other Indicators under Consideration

Graphical illustrations (Figure 2, Figure 3, Figure 4, Figure 5 and Figure 6) based on the obtained results reveal that similar tendencies existed between all types of healthcare costs and GDP by final consumption expenditure—estimated as a percentage. Therefore, the rising GDP led to higher healthcare costs in absolute terms, and we believe that rising GDP was the main factor associated with cost growth in Bulgaria during 2014–2019. The number of hospitalized patients and citizens in retirement age remained constant during the study period; therefore, their impact on rising costs was likely negligible.

## 4. Discussion

Our study findings confirm a positive correlation between economic growth and rising healthcare expenditure. With the increase in GDP, the public healthcare expenditures saw an increase during the study period. A two-way relationship between healthcare system and economic indicators was established. The positive economic situation affects the health system in financial terms, and the development of the health system itself consumes more economic resources. Those relationships are mutual and simultaneous. A study in the USA also determined a positive association between healthcare spending and the economic indicators of labor productivity, per capita GDP, personal income, and other spending, thus confirming the importance of healthcare as an investment in productivity [5].

The growing proportion of elderly people and negative birth rate in the country will require more financial and human resources in the healthcare system and, therefore, will have an impact on expenditures increasing. However, the observed increase in investment in new health technologies, diagnostics, and medical devices might have an offsetting effect in the medium- and long-term period, as it should reduce the cost of treatment and number of chronically ill people for which more public spending should be required [26,27].

Increased life expectancy leads to more spending, because of those additional years in worse health, as well as the higher number of people with health problems or chronic diseases [28,29]. We found a small increase in life expectancy in Bulgaria, which contributed to the increasing number of hospitalizations and total costs covering outpatient and hospital treatment.

A study in Lithuania found that healthcare costs due to hospitalization and reimbursed medicines increased with patient age. Out of the total expenditure, 51.54% was consumed for inpatient treatment, and 30.90% was consumed for reimbursed medicines within the age group of 65–84 years [18]. Our review revealed a negative birth rate of (−0.7%) and a high population over working age (about 1.7 million) in Bulgaria. The aging population is expected to be one of the major consumers of healthcare services due to multimorbidity and increasing need for medicines. Similar results were reported by a population-based, retrospective cohort study in Ontario, Canada where the total costs of multimorbidity varied from 377 to 2073 CAD for young people and from 1026 to 3831 CAD for older patients. The relationship between the degree of multimorbidity and healthcare expenditure depended on age, sex, and income in both age groups [16,17]. The number of emergency department visits and hospitalizations increases for patients with multimorbidity, and healthcare costs increase toward end of life [30].

US healthcare spending increased from 1996 to 2013 mainly as a result of increases in healthcare service price and intensity. The population increase was associated with a 231% or 2695 billion USD spending increase, while aging of the population was associated with a 116% or 1357 billion USD spending increase [31]. In contrast with these findings, our study illustrated that total costs for outpatient and hospital services increased on average by 4.7% and 5.2%, respectively, despite the negative birth rate and decreasing aging population.

The number of hospitalized patients due to cancer diseases in Bulgaria increased annually, while that of diseases of the circulatory system and respiratory system remained almost the same. There is a lack of detailed registries for cardiovascular disease, stroke, type 2 diabetes, and their modifiable risk factors in Bulgaria; therefore, we could not evaluate the advances in therapy [32]. High mortality rate is an indicator of high medicine utilization and hospital admission rate. The increasing number of oncology hospitalizations revealed a growth in oncology medicine utilization and availability, as well as increased expenditures for both reimbursement and hospitalization.

The WHO report revealed that cancer medicines prices are not affordable by healthcare systems globally. Spending on cancer medicines outpaced the growth of overall healthcare expenditure, thus affecting the availability of cancer medicines, especially in low-income countries [33]. On the basis of this finding, we can assume that there is a risk of insufficient financing for the Bulgarian healthcare system.

The amount of healthcare spending also depends on certain risk factors, such as age and sex, with the largest share of spending being for patients aged 65 years and older. Five risk factors were extracted in the USA: high body mass index, high systolic blood pressure, high fasting plasma glucose, dietary risks, and tobacco smoking [34].

A study in Romania exploring health expenditure during 2009–2019 revealed an almost 5–15-fold increase in 5 years (after 2015), except for dentistry and multifunctional medical centers expenditures, where the increase was about 1–2-fold [35]. High-income countries with a higher value of GDP per capita spend more on healthcare in absolute terms [36], thus covering expensive technology and treatment due to an ageing population [37]. One study revealed a positive correlation between the growth of total health expenditure as a percentage of GDP and out-of-pocket spending in Albania, North Macedonia, Bulgaria, Republic of Serbia, and the Russian Federation, whereas a negative correlation was found in Romania, Montenegro, Turkey, Croatia, Estonia, and Greece. The assessment of the relationship between total health expenditure as a percentage of GDP expenditure and GDP per capita revealed significant correlation in 10 of the 15 countries observed, thus confirming that increases in total health expenditure most often comply with GDP growth [38].

Over the study period, GDP per capita increased significantly, whereas extreme inflation was not registered. Final medicines prices are based on specific rules and external referencing mechanisms, but internal situations can also affect them. The prices of medicinal products and healthcare services decreased in 2014 and 2015.Our results show that total cost for outpatient services, total cost for hospital services, and total cost for retail trade, drugstores, optical stores, and other services increased parallel to GDP by final consumption expenditure which could be a result of the aging population or the increased number of specialist visits or hospitalizations. The rising average life expectancy during the study period correlated positively with a high rate of healthcare recourse utilization. NHIF expenditure for medical devices, medicinal products, and dietary foods was similar after 2017. This can be explained by stable medicines prices and the implementation of cost controlling measures in Bulgaria in this period.

The fall in mortality among the elderly population may be explained by a large number of factors. The treatment of diseases that are the most common causes of death such as heart disease and stroke, new therapies for cancer (which is the second most frequent cause of death worldwide), socioeconomic conditions, access to medical care and medicinal products, nutrition, sanitation, community education, and preventative health programs altogether play a significant role in this process [39,40,41].

The limitation of our study is that we focused on selected economic indicators previously explored in literature sources instead of all indicators available in the statistic databases, due to their unstructured datasets. This limited the possibility to explore more tendencies and their impact on healthcare expenditure. An assessment of further indicators [42] could provide a broader view of the relationship between healthcare costs and the factors affecting them.

The increasing life expectancy, aging population, and high number of hospitalized patients found in our study generate high healthcare expenditure. The GDP was the only indicator with a highest impact on healthcare costs, which suggests high resource utilization. Other economic factors probably influence the patient’s treatment and access to therapy and diagnostics. The study findings highlight the need for health system reorganization to specifically cover the treatment of aging people and development of prevention programs. Further studies are needed to explore in more detail the linkage of healthcare costs and economic, demographic, and healthcare system performance indicators over a long-term period. We hope that our study will have practical implications in reorganizing the way in which the economic and healthcare data are compared and analyzed in the country. We can recommend to health authorities to increase the GDP spending on health, and to introduce specific indicators that measure the effect of the healthcare expenditures on income and overall wellbeing of the population.

## 5. Conclusions

Public healthcare expenditure in Bulgaria increased during 2014–2019. The most notable factor was the increase in GDP, suggesting that further economic growth could be associated with higher healthcare costs and medicine affordability in the country. Population aging, average life expectancy, and patient morbidity and hospitalization rate altogether impacted healthcare costs, mainly due to the multimorbidity of the older people and the rising need for outpatient and hospital services, as well as medications.

## Figures and Tables

**Figure 1 healthcare-10-00274-f001:**
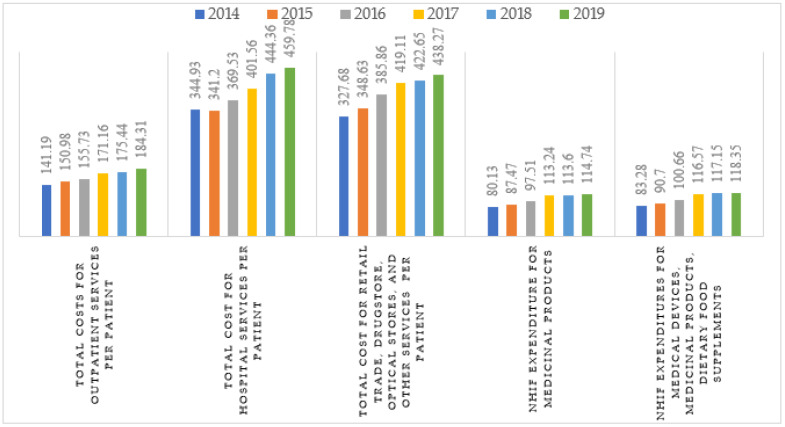
Per patient cost for healthcare services during 2014–2019.

**Figure 2 healthcare-10-00274-f002:**
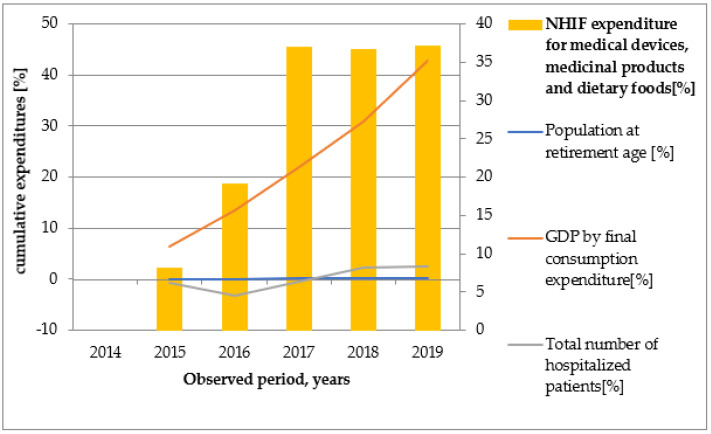
The dependence of NHIF expenditure for medical devices, medicinal products, and dietary foods on GDP by final consumption expenditure, population at retirement age, and number of hospitalized patients.

**Figure 3 healthcare-10-00274-f003:**
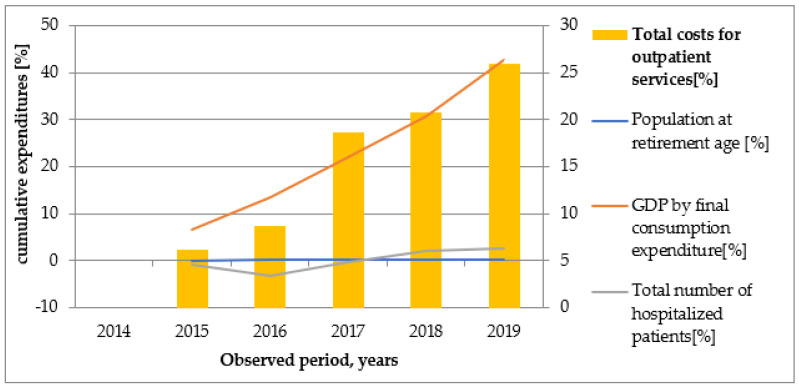
The dependence of total costs for outpatient services on population at retirement age, GDP by final consumption expenditure, and number of hospitalized patients for the investigated period.

**Figure 4 healthcare-10-00274-f004:**
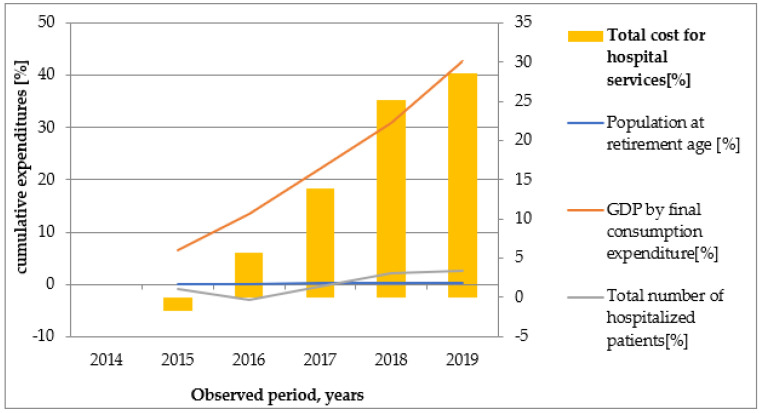
The dependence of total costs for hospital services on population at retirement age, GDP by final consumption expenditure, and number of hospitalized patients.

**Figure 5 healthcare-10-00274-f005:**
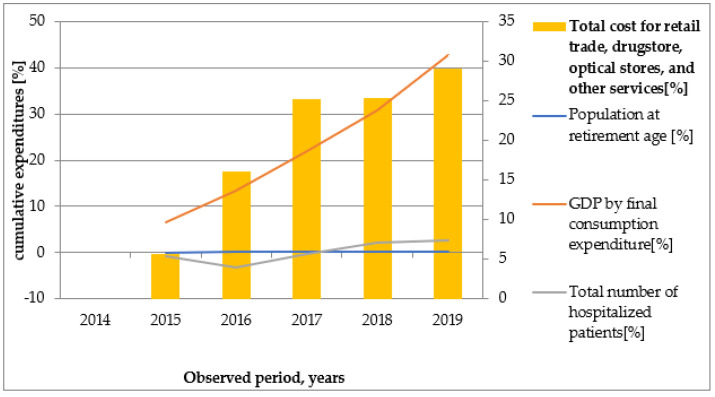
The dependence of total cost for retail trade, drugstores, optical stores, and other services depending on population at retirement age, GDP by final consumption expenditure, and number of hospitalized patients.

**Figure 6 healthcare-10-00274-f006:**
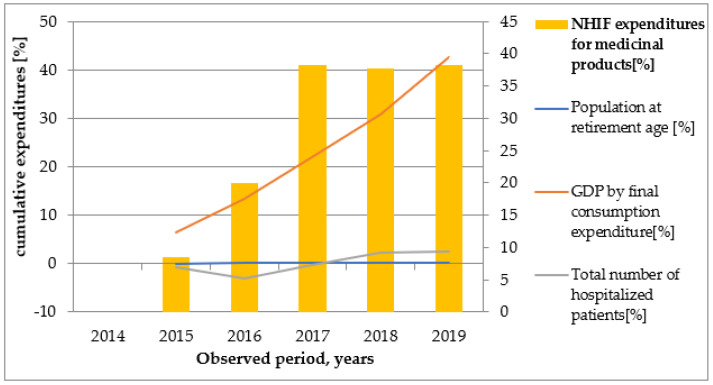
The dependence of NHIF expenditures for medicinal products on population at retirement age, GDP by final consumption expenditure, and number of hospitalized patients.

**Table 2 healthcare-10-00274-t002:** Overview of healthcare expenditure during 2014–2019 (millions of BGN).

	2014	2015	2016	2017	2018	2019	Mean ± SD
Total cost for outpatient services	1017	1080	1105	1207	1228	1281	1153 ± 101
Total cost for hospital services	2484	2441	2624	2831	3111	3196	2781 ± 320
Total cost for retail trade, drugstores, optical stores, and other services	2360	2494	2740	2955	2959	3047	2759 ± 279
NHIF expenditures for medicinal products	577	626	693	798	795	798	714.5 ± 97
NHIF expenditure for medical devices, medicinal products, and dietary foods	600	649	715	822	820	823	738.2 ± 98

**Table 3 healthcare-10-00274-t003:** Yearly expenditure indices as a percentage change in comparison with the previous year.

Differences in Percentage Considering	2014/2015	2015/2016	2016/2017	2017/2018	2018/2019	Mean ± SD
Total costs for outpatient services	6.213	2.397	9.107	1.775	4.328	4.76 ± 2.98
Total cost for retail trade, drugstores, optical stores, and other services	5.681	9.875	7.824	0.128	2.979	5.29 ± 3.85
Total cost for hospital services	−1.747	7.518	7.874	9.875	2.753	5.25 ± 4.71
NHIF expenditures for medicinal products	8.430	10.668	15.277	−0.392	0.307	6.85 ± 6.77

**Table 4 healthcare-10-00274-t004:** Healthcare system performance indicators.

Year	2014	2015	2016	2017	2018	2019
Total number of hospitalized patients	2,238,478	2,217,476	2,169,645	2,220,161	2,288,887	2,292,167
Number of hospitalized patients due to diseases of the circulatory system [22]	321,074	317,486	324,178	328,928	325,543	327,676
Number of hospitalized patients due to respiratory system diseases	239,978	223,661	212,770	218,498	222,124	229,217
Number of hospitalized patients due to cancer diseases	159,613	156,995	165,502	167,282	182,252	172,549
Mortality from the diseases of the circulatory system	71,760	72,028	70,459	71,997	70,546	69,632
Mortality from the oncology diseases	18,113	18,020	17,294	17,429	17,462	18,298
Mortality in infants including neonates (0–1 years of age)	517	434	423	408	358	342

**Table 5 healthcare-10-00274-t005:** Demographic characteristics of population published by NSI.

Parameter	2014	2015	2016	2017	2018	2019
Population size	7,202,198	7,153,784	7,101,859	7,050,034	7,000,039	6,951,482
Average life expectancy	74.5	74.7	74.9	74.8	75.0	74.9 *
Population over working age (retirement age)	1,734,089	1,740,749	1,734,718	1,735,538	1,732,018	1,728,730
Ratio of the population aged 65 and over to the population aged 15 to 64	30.2	31.1	31.8	32.5	33.2	33.8

* Specific data are not published; average life expectancy published for the period 2017–2019 was 74.9 years.

**Table 6 healthcare-10-00274-t006:** Economic indicators in Bulgaria during 2014–2019 (BGN).

Year	2014	2015	2016	2017	2018	2019	Mean ± SD
GDP by final consumption expenditure (million BGN)	83,885	89,362	95,131	102,345	109,743	119,772	100,039.7 ± 13,326.03
GDP per capita	12,800	13,186	13,901	14,687	15,534	16,508	14,436 ± 1420.3
Average monetary expenditures per person	4443	4605	4699	5161	5710	6158	5129.333 ± 682.22
Average monetary expenditures for healthcare per person	235	248	263	280	313	386	287.5 ± 55.36
Gross monetary income per person	4591	4741	4945	5288	5691	6279	5255.83 ± 638.90
Average yearly inflation in healthcare sector (%)	−3.4	−1.6	−0.2	0.2	0.2	1.3	−0.58 ± 1.67
Inflation of medicinal product prices (%)	−1.6	−2.3	−0.5	0.0	0.1	1.3	−0.5 ± 1.29
Inflation of medicinal devices and equipment prices (%)	1.3	0.8	0.8	2.1	1.5	1.1	1.27 ± 0.49

## Data Availability

The data that support the findings of this study are available on request from the corresponding author. Some of the data is available published on the official websites of the state bodies: www.nsi.bg/bg/content/14521 (accessed on 20 January 2022); https://www.nhif.bg/page/2035 (accessed on 20 January 2022), and https://ncpha.government.bg/index/124-spravochnik-zdraveopazvane.html (accessed on 20 January 2022).

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
