# Peer review of "Analysis of Healthcare Expenditures in Bulgaria"

_healthcare, 2022, doi:10.3390/healthcare10020274_

Round 1
Reviewer 1 Report
I would like to thank the authors for this research that aims to review the tendencies in public healthcare expenditures in Bulgaria and to analyze the influence of the demographic, economic, and healthcare system capacity indicators on expenditures dynamics. The outcomes of this research evaluated the main factors driving public expenditure in Bulgaria.
The research subject is timely and highly interesting especially for ageing countries like Bulgaria. So, the research subject is interesting and timely.
The research still needs several improvements on different parts.
The abstract needs to be redesigned in order to Cleary show the results and implications of the research.
The authors omitted to perform a literature review which Greatly affected the quality of the research. I recommend to authors to either extend their introduction or add a literature review section. This will provide more value to their research, make it stronger, and establish links with previous studies.
Still, there are a lot of grammatical errors and the whole manuscript needs deep proofreading by a native speaker.
At the end of your research, it is important to add the limitations, implications, and recommendations of your research. In fact, we need to understand the importance of your research.
Several other comments are directly attached to the manuscript and could help authors to improve their research.

Author Response
Dear reviewer,
Thank you very much!
Your support and recommendations were very important to improve quality of the study.
Regards,
Zornitsa

Reviewer 2 Report
After reading the paper I think that it provides a valuable contribution especially regarding the structure of the health care system in Bulgaria. The authors investigate three main aspects affecting public health care expenditure: demographic characteristics, economic factors and performance.
However, I would recommend the publication of the paper only after the revision of the following aspects.
1) The author claim to use a nonparametric methodology to test statistically significant differences between healthcare expenditures during the observed period, based on Mann-Whitney. First I think that some more details and references regarding this methodology should be included in the paper. However, my most important concern is that no formal tests are presented in the paper. The analysis is based only on graphs. So I would suggest either including formal tests or changing the methodological approach stating since the beginning that the paper is based exclusively on graphs.
2) I would suggest adding a brief summary of the structure of the paper at the end of the introduction, allowing the reader to know the contents of each paragraph.
3) Among the health care system indicators I would suggest the inclusion of neonatal and/or infant mortality. Please refer to Porcelli 2014 for the usefulness of those indicators for the measurement of performance.
4) To improve the readability of figures I would include different scales for the variables plotted, adding a second vertical axis.
5) Overall the editing of the text must be improved many times words without space appear in the text.
References
Porcelli F. 2014. “Electoral Accountability and Local Government Efficiency: Quasi-Experimental Evidence from the Italian Health Care Sector Reforms”, The Economics of Governance 15(3):221-251.
Author Response

(The authors gave the same response as above.)

Round 2
Reviewer 1 Report
I would like to thank the authors for this research that aims to review the tendencies in public healthcare expenditures in Bulgaria and to analyze the influence of the demographic, economic, and healthcare system capacity indicators on expenditures dynamics. The outcomes of this research evaluated the main factors driving public expenditure in Bulgaria.
The research subject is timely and highly interesting especially for ageing countries like Bulgaria. So, the research subject is interesting and timely.
The researchers made the necessary changes as suggested in the previous round.
Congratulations for them and their efforts are valued.
Author Response
Dear reviewer,
Thank you very much for all the valuable comments.
Reviewer 2 Report
After the first revision, the paper is improved, most of my previous suggestions have been included. However, I think that some minor changes are still necessary before publication. In particular, I would suggest checking the editing of figures in tables 4, 5 and 6 since sometimes there is no coherent use of dots and commas and there are spaces between figures. Finally, in figures 2, 3, 4, 5 and 6 I would recommend reporting on a second vertical axis the percentage of the population at retirement age and the total number of hospitalized patients so as to improve the readability of the graphs.
Author Response
Dear reviewer,
Thank you very much for all the valuable comments. We have revised all tables and the included figures are corrected. We would like to be thankful especially for the advice for graphical revision as the change of their type makes the illustration much more interesting and easier for the readers. In this relation, we can confirm that a second vertical axis is added. Using the different types of graphics we found out that when the main explored factors used as a basis are included in the second vertical axis, the presented data is more illustrative and in fact the figure conception can be presented in the best way. We sincerely hope that this illustration will be the best from your own point of view. If another revision is needed we are ready to prepare it immediately.

This manuscript is a resubmission of an earlier submission. The following is a list of the peer review reports and author responses from that submission.
Round 1
Reviewer 1 Report
Authors focus on analysis of healthcare expenditures in Bulgaria.
There are 8 authors, although the paper is very short. Lack of any original approach. Lack of research work. There is no description of research methodology. The paper covers just descriptive statistics presentation.
Reviewer 2 Report
I would like to thank the authors for this research that aims to review the tendencies in public healthcare expenditures in Bulgaria and to analyze the influence of the demographic, economic and healthcare system capacity indicators on expenditures dynamics. The outcomes of this research evaluated the main factors driving public expenditure in Bulgaria.
The research subject is timely and highly interesting especially for ageing countries like Bulgaria. So, the research subject is interesting, timely, however, it is not perfectly aligned with aim and scope of the Healthcare journal.
The title doesn’t reflect the content of the research and needs to focus more on the analysis of Healthcare expenditures in Bulgaria.
Authors omitted to perform a literature review which Greatly affected the quality of the research. Even the references and numbers cited in the introduction are not accurate or were note accurately reported. Example, percentages cited in lines 29, 31, 33, 37 do not have any sense and are completely wrong.
There are lot of grammatical errors and the whole manuscript needs deep proofreading by a native speaker.
Tables are difficult to read, numbers are long and need to be simplified (by millions for example).
Discussion section is poor. In this section you need to discuss your results and link with previous research. The actual manuscript uses this section as literature review section. Once you cite something you need to link it with your results and not simply citing it.
Several other comments are directly attached to the manuscript and could help authors to improve their research.